# Perspective: Summary Statistics of Learning[*]

**Jacob A. Zavatone-Veth**[1,2], **Blake Bordelon**[3], and **Cengiz Pehlevan**[1,4,5]
[1]Center for Brain Science, [2]Society of Fellows,
[3]Center for Mathematical Sciences and Applications,
[4]John A. Paulson School of Engineering and Applied Sciences,
[5]Kempner Institute for the Study of Natural and Artificial Intelligence,
Harvard University, Cambridge, MA, USA
{jzavatoneveth@fas, blake_bordelon@g, cpehlevan@seas}.harvard.edu

**Editors:** Marco Fumero, Clementine Domine, Zorah Lähner, Irene Cannistraci, Bo Zhao, Alex Williams

## Abstract

How can we make sense of large-scale recordings of neural activity across learning? Theories of neural network learning with their origins in statistical physics offer a potential answer: for a given task, there are often a small set of summary statistics that are sufficient to predict performance as the network learns. Here, we review recent advances in how summary statistics can be used to build theoretical understanding of neural network learning. We then argue for how this perspective can inform the analysis of neural data, enabling better understanding of learning in biological and artificial neural networks.

## 1 Introduction

Experience reshapes neural population activity, molding an animal's representations of the world as it learns to perform new tasks. Thanks to advances in experimental technologies, it is just now becoming possible to measure changes in the activity of large neural populations across the course of learning [2–8]. However, with this new capability comes the challenge of identifying which features of high-dimensional activity patterns are meaningful for understanding learning. While analyses of representations have begun how to elucidate how learning reshapes the structure of activity, it is not in general clear whether these measurements are sufficient to understand how representational changes relate to behavior [4, 9–11].

In this Perspective, we propose that the principled identification of **summary statistics of learning** offers a possible path forward. This framework is grounded in theories of the statistical physics of learning in neural networks, which show that low-dimensional summary statistics are often sufficient to predict task performance over the course of learning [12–14]. We argue that thinking systematically about summary statistics gives new insight into what existing approaches of quantifying neural representations reveal about learning, and allows identification of what additional measurements would be required to constrain models of plasticity. We emphasize that the goal of this Perspective is not to advocate for the use of a particular set of summary statistics, but rather to explain the general philosophy of this approach to understanding learning in high dimensions.

---

[*]This extended abstract is an abbreviated version of our long-form perspective [1].

Proceedings of the III edition of the Workshop on Unifying Representations in Neural Models (UniReps 2025).

## 2 What is a summary statistic?

We posit that summary statistics of learning must satisfy two minimal desiderata: First, **they must be low-dimensional.** That is, their dimension is low relative to the number of neurons in the network of interest. Indeed, most summary statistics we will encounter are determined by averages over the population of neurons. Second, **they must be sufficient to predict behavior across learning.** From a theoretical standpoint, there should exist a closed set of equations describing the evolution of the summary statistics that predict the network's performance. Summary statistics satisfying these two desiderata are often highly interpretable thanks to their clear relationship to the network architecture and learning task (Section 3). However, the summary statistics relevant for predicting performance may not be sufficient to predict all statistical properties of population activity (Section 4).

## 3 Summary statistics in theories of neural network learning

We now review how summary statistics emerge naturally in theoretical analyses of neural network learning. Out of many theoretical results, we focus on two example settings: here we discuss batch learning in wide and deep networks [12–25], and in Appendix A we discuss online learning from high-dimensional data in shallow networks. These model problems illustrate how relevant summary statistics may be identified given a task, network architecture, and learning rule.

Consider a deep fully-connected network with input $\mathbf{x} \in \mathbb{R}^D$, at training time $t$:

$$f(\mathbf{x}, t) = \frac{1}{\gamma\sqrt{N}} \sum_{i=1}^{N} w_i(t)\phi(h_i^{(L)}(\mathbf{x}, t)),$$

$$h_i^{(\ell+1)}(\mathbf{x}, t) = \frac{1}{\sqrt{N}} \sum_{j=1}^{N} W_{ij}^{(\ell)}(t)\phi(h_j^{(\ell)}(\mathbf{x}, t)), \quad \ell \in \{1, \dots, L+1\},$$

$$h_i^{(1)}(\mathbf{x}, t) = \frac{1}{\sqrt{D}} \sum_{j=1}^{D} W_{ij}^{(0)}(t)x_j.$$

Suppose we use gradient flow to minimize the average error on a fixed set of training examples, and consider a regime where the hidden layer width $N$ is small relative to the input dimension $D$ (Figure 1a). What are the relevant summary statistics? Applying the chain rule, one finds that

$$\frac{d}{dt}f(\mathbf{x}, t) = -\mathbb{E}_{\mathbf{x}'} \sum_{\ell} G^{(\ell+1)}(\mathbf{x}, \mathbf{x}', t, t)\Phi^{(\ell)}(\mathbf{x}, \mathbf{x}', t, t)\frac{\partial\mathcal{L}}{\partial f(\mathbf{x}', t)},$$

where $\mathcal{L}$ is the loss function and $\mathbb{E}_{\mathbf{x}'}$ denotes expectation over the training dataset [20, 26, 27]. Here,

$$\Phi^{(\ell)}(\mathbf{x}, \mathbf{x}', t, t') = \frac{1}{N} \sum_{i=1}^{N} \phi(h_i^{(\ell)}(\mathbf{x}, t))\phi(h_i^{(\ell)}(\mathbf{x}', t'))$$

are **representational similarity matrices**, and

$$G^{(\ell)}(\mathbf{x}, \mathbf{x}', t, t') = \frac{1}{N} \sum_{i=1}^{N} g_i^{(\ell)}(\mathbf{x}, t)g_i^{(\ell)}(\mathbf{x}', t'), \quad g_i^{(\ell)}(\mathbf{x}, t) \equiv \gamma\sqrt{N}\frac{\partial f(\mathbf{x}, t)}{\partial h_i^{(\ell)}(\mathbf{x}, t)},$$

are **gradient similarity matrices**, which at each layer compare the hidden states and the gradient signals for each pair of data points and each pair of training times. Thus, as these matrices determine the dynamics of $f$, they are suitable summary statistics if they are low-dimensional relative to the set of synaptic weights, and if we can write down a closed set of equations for their dynamics.

First, it is easy to see that the criterion of dimensionality reduction requires that the number of training examples $P$ is much less than the network width $N$, as the number of similarity matrix elements and the number of synaptic weights are of order $P^2$ and $N^2$, respectively. Second, it turns out that one can close the equations for $\Phi^{(\ell)}$ and $G^{(\ell)}$ provided that the width is large and that the synaptic weights start from an uninformed initial condition (*i.e.*, Gaussian random matrices) [20, 26–28]. Depending on how weights and learning rates are scaled, one can obtain different types of large-width ($N \to \infty$) limits (Figure 1b). In the *lazy / kernel* limit where $\gamma$ is constant, these representational similarity matrices are static over the course of learning [26, 27]. If instead $\gamma \propto \sqrt{N}$), these objects evolve in a task-dependent manner even as $N \to \infty$ (Figure 1c) [20, 28].

A significant line of recent work in neuroscience aims to quantify neural representations and compare them across networks through analysis of representational similarity matrices [10, 11, 29, 30]. Here, we see that these kernel matrices arise naturally as summary statistics of forward signal propagation in wide and deep neural networks (Figure 1c-d). At the same time, those results show that tracking only feature kernels is *not* in general sufficient to predict performance over the course of learning. One needs access also to coarse-grained information about the plasticity rule in the form of gradient kernels, and to information about the network outputs (for instance $\partial\mathcal{L}/\partial f$). More theoretical work

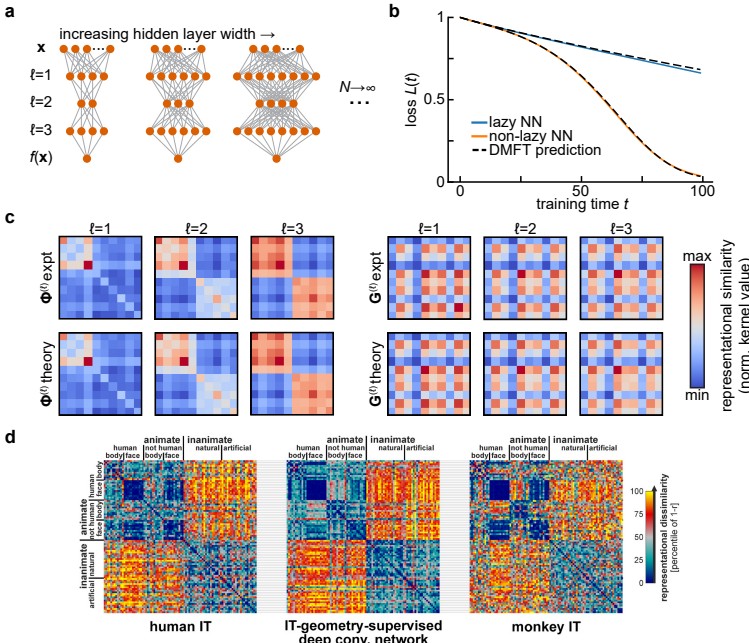

Figure 1: **Representational similarity kernels in wide neural network models and in the brain.**
**a**. Diagram of the infinite-width limit of a deep feedforward neural network. For a fixed input
and output dimension, one considers a sequence of networks of increasing hidden layer widths.
**b**. Predicting the performance of width-2500 fully-connected networks with three hidden layers
and tanh activations over training using the dynamical mean-field theory described in Section 3.
Networks are trained on a synthetic binary classification dataset of 10 examples, with 5 examples
assigned each class at random. This leads to block structure in the final representations. Adapted
from Bordelon and Pehlevan [20]. **c**. The summary statistics in the dynamical mean field theory for
the network in **b** are representational similarity kernels ($\mathbf{\Phi}^{(\ell)}$; *left*) and gradient similarity kernels
($\mathbf{G}^{\ell}$; *right*) for each layer. The top row shows kernels estimated from gradient descent training, and
the bottom row the theoretical predictions. All kernels are shown at the end of training ($t = 100$).
Adapted from Bordelon and Pehlevan [20]. **d**. Comparing representational similarity kernels across
models and brains. Here, similarity is measured using the Pearson correlation $r$, and the *dissimilarity*
$1 - r$ is plotted as a heatmap. Kernels resulting from fMRI measurements of human inferior temporal
(IT) cortex (*left*) and electrophysiological measurements of macaque monkey IT cortex (*right*) are
compared with the kernel for features from a deep convolutional neural network after optimal re-
weighting to match human IT (*center*). Adapted from Figure 10 of Khaligh-Razavi and Kriegeskorte
[31] with permission from N. Kriegeskorte under a CC-BY License.

is required to determine how to reliably estimate these gradient kernels from data, thereby providing
a means to gain coarse-grained information about the underlying plasticity rule.

## 4   Implications for neural measurements

The two example settings detailed in Section 3 show how the relevant summary statistics of learning
depend on network architecture and learning rule. Theoretical studies are just beginning to map out the
full space of possible summary statistics for different network architectures [13–25]. Though details
of the relevant summary statistics vary depending on the scaling regime and task—as illustrated by
the examples above, where network width, training dataset size, and learning rule change the relevant
statistics and their effective dynamics—they share broad structural principles. Thanks to these
common structural features, these varied theories of summary statistics have common implications
for the analysis and interpretation of neuroscience experiments.

## 4.1 Benign subsampling

The summary statistics encountered in Section 3 are robust to subsampling thanks to their basic nature as averages over the population of neurons. These statistical theories in fact post a far stronger notion of benign subsampling: they result in neurons that are statistically exchangeable. This is highly advantageous from the perspective of long-term recordings of neural activity, as reliable measurement of summary statistics does not require one to track the exact same neurons over time. Instead, it suffices to measure a sufficiently large subpopulation on any given day. This obviates many of the challenges presented by tracking neurons over multiple recording sessions [2]. Moreover, the variability and bias introduced by estimating summary statistics from a limited subset of relevant neurons can be characterized systematically [32–34]. Taken together, these properties mean that summary statistics are relatively easy to estimate given limited neural measurements, provided that exchangability is not too strongly violated [35]. There are limits, however, to how far one can subsample. For instance, representational similarity kernels are more affected by small, coordinated changes in the tuning of many neurons than large changes in single-neuron tuning (Figure 2) [4]. Determining the minimum number of neurons one must record in order to predict generalization dynamics across learning will be an important subject for future theoretical work [4, 35].

## 4.2 Invariances and representational drift

Though by our definition the summary statistics mentioned in Section 3 are sufficient to predict the network's performance, they are not sufficient statistics for all properties of the neural code. In particular, in part because they arise from theories in which neurons become exchangable, they have many invariances. These invariances mean that individual tuning curves can change substantially without altering the population-level computation [4]. For instance, the representational similarity kernels are invariant under rotation of the neural code at each layer. Similarly, overlaps with task-relevant directions are invariant to changes in the null space of those low-dimensional projections. These invariances mean that focusing on summary statistics of learning sets a particular aperture on what aspects of representations one can assay.

At the same time, the invariances of summary statistics have important consequences for functional robustness. In particular, they are closely related to theories of representational drift, the seemingly puzzling phenomenon of continuing changes in neural representations of task-relevant variables despite stable behavioral performance [2, 36]. Many models of drift explicitly propose that representational changes are structured in such a way that certain summary statistics are preserved (Figure 2a) [2, 37, 38]. Identifying the invariances of the summary statistics sufficient to determine task performance can allow for a systematic characterization of what forms of drift can be accommodated by a given network. Conversely, identifying the invariances of a representation once task performance stabilizes might suggest which summary statistics are relevant for the learning problem at hand.

## 4.3 Universality

An important lesson from the theory of high-dimensional statistics is that of *universality*: certain coarse-grained statistics are asymptotically insensitive to the details of the distribution. The most prominent example is the central limit theorem: the distribution of the sample mean of independent random variables tends to a Gaussian as the number of samples becomes large. A broader class of universality principles arise in random matrix theory: the distribution of eigenvalues and eigenvectors of a random matrix often become insensitive to details of the distribution of the elements as the matrix becomes large. Most famously, the Marčenko-Pastur theorem specifies that the singular values of a matrix with independent elements have a distribution that depends only on the mean and variance of the elements [39]. In learning problems, universality manifests through insensitivity of the model performance to details of the distributions of parameters or of features [40, 41].

From the perspective of summary statistics, statistical universality can allow simple theories to make informative macroscopic predictions even if they do not capture detailed properties of single neurons. For instance, the mean-field description of the learning dynamics of wide neural networks introduced in Section 3 are universal in that they depend on the initial distribution of hidden layer weights only through its mean and variance, even though the details of that distribution will affect the distribution of weights throughout training (Figure 2b-d) [42, 43]. Like the invariances to transformations of the neural population code mentioned before, this is nonetheless a double-edged sword: these universality properties mean that focusing on predicting performance commits one to coarse-graining away certain

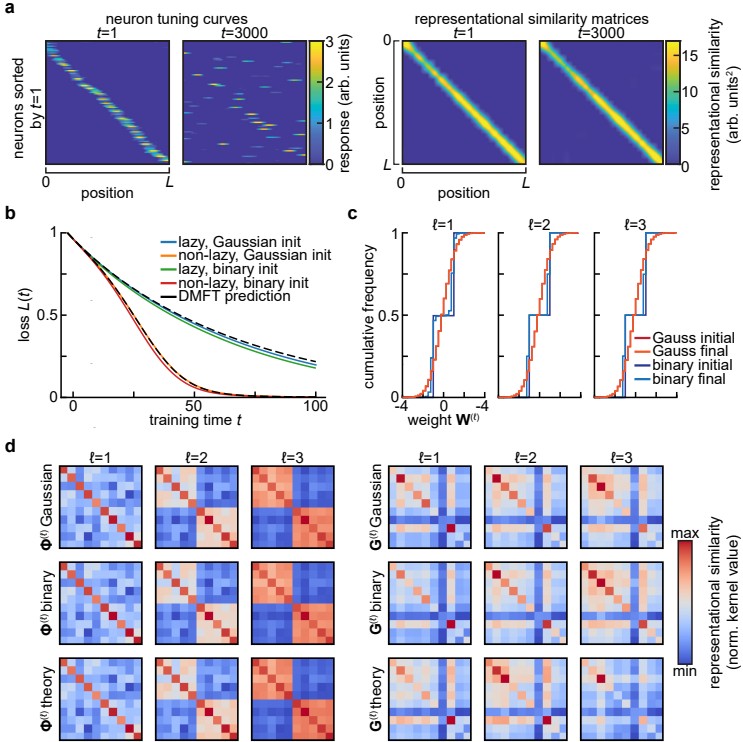

Figure 2: **Invariance and universality in summary statistics.**
**a**. Stable summary statistics despite drifting single-neuron responses. In Qin et al. [38]'s model of representational drift, single neurons are strongly tuned to a spatial variable, yet their tuning changes dramatically over time (*left*). Despite this drift, the similarity of the population representations of different spatial positions remains nearly constant (*right*). Adapted from Figure 5e of Qin et al. [38].
**b**. Universality of summary statistics in wide and deep networks with respect to the distribution of initial weights. Setting is as in Figure 1b-c, but also including a network for which the weights are initially drawn from $\{-1, +1\}$ with equal probability. Here, $N = 2000$, and a different realization of the random task is sampled relative to Figure 1b-c, so the loss curves are not identical. **c**. Cumulative distribution of weights at the start (*initial*) and end (*final*) of training for the networks shown in (**b**). Note that the small change in the weight distributions for the Gaussian-initialized networks is not visible at this resolution, and that one expects the size of weight changes to scale with $1/\sqrt{N}$ [20]. **d**. Feature and gradient kernels at the end of training for the networks in **b**. No substantial differences are visible between networks initialized with different weight distributions.

microscopic aspects of neural activity. Though these features are not required to predict macroscopic behavior, they may be important for understanding biological mechanisms.

## 5   Discussion

As reviewed here, the core insight of the statistical mechanics of learning in neural networks is the existence of low-dimensional summary statistics sufficient to predict behavioral performance. We now conclude by discussing future directions for theoretical inquiry. The models reviewed here are composed of exchangable neurons, which simplifies the relevant summary statistics and renders them particularly robust to subsampling. However, the brain has rich structure that can affect which summary statistics are sufficient to track learning and how those summary statistics may be measured. Biological neural networks are embedded in space, and their connectivity and selectivity is shaped by spatial structure [44–46]. Notably, many sensory areas are topographically organized: neurons with similar response properties are spatially proximal [47, 48]. Moreover, neurons can be classified into genetically-identifiable cell types [49], which may play distinct functional roles during learning [3, 50]. Future theoretical work must contend with these biological complexities in order to determine the relevant summary statistics of learning subject to these constraints.

## Acknowledgments and Disclosure of Funding

J.A.Z.-V. is supported by the Office of the Director of the National Institutes of Health under Award Number DP5OD037354. The content is solely the responsibility of the authors and does not necessarily represent the official views of the National Institutes of Health. JAZV is further supported by a Junior Fellowship from the Harvard Society of Fellows. B.B. is supported by a Google PhD Fellowship. C.P. is supported by NSF grant DMS-2134157, NSF CAREER Award IIS-2239780, DARPA grant DIAL-FP-038, a Sloan Research Fellowship, and The William F. Milton Fund from Harvard University. This work has been made possible in part by a gift from the Chan Zuckerberg Initiative Foundation to establish the Kempner Institute for the Study of Natural and Artificial Intelligence.

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

# A Online learning in shallow neural networks with high dimensional data

Classical models of online gradient descent learning in high dimensions can be often be summarized with simple summary statistics [12, 13, 16, 17, 24, 51–53]. In this section, we discuss how the generalization performance of perceptrons and shallow (two-layer) neural networks trained on large quantities of high dimensional data can be summarized by simple weight alignment measures. Most simply, the perceptron model $f(\mathbf{x}) = \sigma\left(\frac{1}{\sqrt{D}}\mathbf{w}\cdot\mathbf{x}\right)$ seeks to learn a weight vector $\mathbf{w} \in \mathbb{R}^D$ which correctly classifies a finite set of randomly sampled training input-output pairs $(\mathbf{x}_\mu, y_\mu)$. If the inputs are random, $\mathbf{x}_\mu \sim \mathcal{N}(0, \mathbf{I}_D)$, and the targets $y_\mu = y(\mathbf{x}_\mu)$ are generated by a **teacher network** $y(\mathbf{x}) = \sigma\left(\frac{1}{\sqrt{D}}\mathbf{w}_\star\cdot\mathbf{x}\right)$, then the generalization performance (performance of the model on new *unseen data*, $\mathbb{E}_\mathbf{x}[(f(\mathbf{x}) - y(\mathbf{x}))^2]$) is completely determined by the overlap of $\mathbf{w}$ with itself and with the target direction $\mathbf{w}_\star$

$$Q = \frac{1}{D}\mathbf{w}\cdot\mathbf{w} \, , \ R = \frac{1}{D}\mathbf{w}\cdot\mathbf{w}_\star. \tag{1}$$

If the learning rate is scaled appropriately with the dimension $D$, the high-dimensional (large-$D$) limit of online stochastic gradient descent is given by a deterministic set of equations for $Q$ and $R$:

$$\frac{d}{d\tau}\begin{bmatrix}Q(\tau)\\R(\tau)\end{bmatrix} = \mathbf{F}[Q(\tau), R(\tau)], \tag{2}$$

where the continuous training 'time' $\tau$ is the ratio of the number of samples seen to the dimension and $\mathbf{F} : \mathbb{R}^2 \to \mathbb{R}^2$ is a nonlinear function that depends on the learning rate, the loss function, and the link function $\sigma(\cdot)$ [13, 16, 17, 24, 51]. Integrating this update equation allows one to predict the evolution of the generalization error as more training data are provided to the algorithm. Despite the infinite dimensionality of the original optimization problem, only two dimensions are necessary to capture the dynamics of generalization error.

The analysis of online perceptron learning can be extended to two layer neural networks with a small number of hidden neurons $N$,

$$f(\mathbf{x}) = \frac{1}{N}\sum_{i=1}^{N} a_i\,\phi\left(h_i(\mathbf{x})\right) \quad h_i(\mathbf{x}) = \frac{1}{\sqrt{D}}\mathbf{w}_i\cdot\mathbf{x} \, , \ i \in \{1, ..., N\}. \tag{3}$$

$$y(\mathbf{x}) = \sigma\left(h_1^\star(\mathbf{x}), ..., h_K^\star(\mathbf{x})\right) \quad h_k^\star(\mathbf{x}) = \frac{1}{\sqrt{D}}\mathbf{w}_k^\star\cdot\mathbf{x} \, , \ k \in \{1, ..., K\}. \tag{4}$$

In this setting with isotropic random data, the relevant summary statistics are the readout weights $\mathbf{a} \in \mathbb{R}^N$, along with **overlap matrices** $\mathbf{Q} \in \mathbb{R}^{N\times N}$ and $\mathbf{R} \in \mathbb{R}^{N\times K}$ with entries

$$Q_{ij} = \frac{1}{D}\mathbf{w}_i\cdot\mathbf{w}_j \, , \ R_{ik} = \frac{1}{D}\mathbf{w}_i\cdot\mathbf{w}_k^\star \tag{5}$$

For this system, we can track the gradient descent dynamics for $\mathbf{a}$, $\mathbf{Q}$, and $\mathbf{R}$ through a generalization of Equation (2) [16, 17, 52, 53]. This reduces the dimensionality of the dynamics from the $N + DN$ trainable parameters $\{a_i\}, \{w_j\}$ to $N + N^2 + NK$ summary statistics, which is significant when $D \gg N + K$. This reduction enables the application of analyses that cannot scale to high dimensions, for instance control-theoretic methods to study optimal learning hyperparameters and curricula [54, 55]. Recent works have also begun to study approximations to these summary statistics when the network width $N$ is also large, as further dimensionality reduction if possible when $\mathbf{Q}$ and $\mathbf{R}$ have stereotyped structures [24, 56].

Under what conditions is this reduction possible? Fundamentally, the summary statistics $\mathbf{a}$, $\mathbf{Q}$, and $\mathbf{R}$ are sufficient to determine the network's performance so long as the preactivations $h_i$ and $h_k^\star$ are approximately Gaussian. Thus, one can relax the assumption that the inputs $\mathbf{x}$ are exactly Gaussian so long as a central limit theorem applies to $h_i$ and $h_k^\star$ [16, 52]. Moreover, one can allow for correlations between the different input dimensions so long as $h_i$ and $h_k^\star$ remain Gaussian. If $\mathbb{E}[\mathbf{x}\mathbf{x}^\top] = \mathbf{\Sigma}$, with a modification of the definition of the overlaps to $Q_{ij} = \frac{1}{D}\mathbf{w}_i\cdot\mathbf{\Sigma}\mathbf{w}_j$ and $R_{ik} = \frac{1}{D}\mathbf{w}_i\cdot\mathbf{\Sigma}\mathbf{w}_k^\star$ a similar reduction applies [24]. One can even consider extensions to plasticity rules other than stochastic gradient descent. For example, online node perturbation leads to a different effective dynamics for the same set of summary statistics [57, 58].

How could the overlaps $\mathbf{Q}$ and $\mathbf{R}$ be accessed from measurements of neural activity? And, in the absence of detailed knowledge of a teacher network, how could one identify the relevant overlaps? Under the simple structural assumptions of these models, one could estimate the overlaps from

covariances of network activity across stimuli, *i.e.*, with isotropic inputs one has $\mathbb{E}_{\mathbf{x}}[h_i h_k^\star] = R_{ik}$ and $\mathbb{E}_{\mathbf{x}}[h_i h_j] = Q_{ij}$. Moreover, one can in some cases detect this underlying low-dimensional structure by examining the principal components of the learning trajectory [15]. However, more theoretical work is required in this vein.

