# OpenReview forum: "Perspective: Summary Statistics of Learning"
_NeurIPS.cc/2025/Workshop/UniReps — UniReps2025_

### Official Review · Reviewer_nZ4Y · 2025-09-12
**Short perspective on how leveraging summary statistics helps to provide a deeper understanding of neural learning dynamics**

**Confidence:** 5

**Review:**

## Summary
This perspective paper promotes using low-dimensional summary statistics inspired by statistical physics to interpret high-dimensional neural activity data during learning in both biological and artificial neural networks. In addition, the paper reviews theoretical foundations, defines key criteria for these statistics, and explores their implications for experimental neuroscience, illustrating how they can predict performance while highlighting gaps in current representational analyses.

## Main contribution of the paper:
- Identifying that low-dimensional summary statistics that are  sufficient to predict behavioral performance across learning, based on theories from neural network learning that are rooted in statistical physics.
- Review specific theoretical examples, such as batch learning in wide and deep networks, in which representational and gradient similarity matrices emerge as key summary statistics.
- Illustrate the consequences for neural measurements, including benign sub-sampling, invariances leading to representational drift, and universality principles that allow coarse-grained predictions insensitive to microscopic details.

## Strengths:
- The paper is clear for general audience, and it bridges between theory and experiments.
- Challenges in long-term neural recordings are highlighted.

## Weaknesses:
- The paper lacks solid examples or case studies applied to real neural data.
- No new tools/methods/algorithms for estimating summary statistics from data were introduced.

**More comments about the main body of the paper:**

++Abstract++
- The abstract needs to be revised as it is a bit short and doesn't fully reflect all of the questions being investigated within the paper.
- In line 6: change 'argue for' with illustrate or show.

++Introduction++
- In line 10: 'molding an animal’s representations of the world' you didn't cite any references to support that, even though there are some studies from a group at NTNU doing similar things on salmon fishes, refer to Martin Føre works [1]-[2]
- In line 15: there's a typo in 'how to elucidate how learning reshapes'
- In lines 15~16: 'it is not in general clear...' it will be better to reformulate it as ''there is a gap in the literature on whether these measurements are sufficient to understand ...''

++What is a summary statistic?++
- In line 28: exchange 'desiderata' with criteria
- In lines 35~35: the last sentence 'summary statistics relevant for predicting performance may not be sufficient': illustrate why shortly before referring to Section 4 similar to what you wrote in the sentence before it

++Summary statistics in theories of neural network learning++
- Since the paper is written in a form of a short survey and at the end you are shedding light on how summary statistics gives insight  to understand how neural representations reflect learning processes, then its better to define all of the symbols and mathematical notations in the main body of the paper for the general reader who's not into the field.
- In relation to the previous comment, revise how the mathematical notations are defined in the Appendix
- Typo in line 57: starting instead of 'star'


++References++

[1] Jónsdóttir, Kristbjörg Edda, et al. "Detection of a stress related acoustic signature by passive acoustic monitoring in Atlantic salmon farming." Aquacultural Engineering 107 (2024): 102472.

[2] Bloecher, Nina, et al. "Assessment of activity and heart rate as indicators for acute stress in Atlantic salmon." Aquaculture International 32.4 (2024): 4933-4953.

**Score:**

2

**Topic Fit:**

2

---

### Official Review · Reviewer_MBRc · 2025-09-15
**An interesting perspective arguing that low-dimensional summary statistics, grounded in statistical physics, can bridge theory and neural data analysis of learning.**

**Confidence:** 3

**Review:**

This extended abstract presents a perspective on summary statistics of learning, drawing on statistical physics to argue that low-dimensional descriptors can often predict learning dynamics in neural networks. The authors propose that such summary statistics may also provide a principled way to analyze large-scale neural recordings across learning. The work is clearly written, and makes a valuable cross-disciplinary connection between machine learning theory and systems neuroscience. While some sections are technically dense, the overall argument is accessible and has potential to inspire interesting discussions. The piece succeeds in opening a conceptual discussion about which aspects of neural data matter for understanding learning, how invariances shape interpretation, and how subsampling affects analysis. It also discusses limitations (e.g. assuming interchangeability of neurons) and outlines future directions and open research questions (e.g. how large a population is needed to compute summary statistics).

Strengths:

Well-structured and clearly defines “summary statistics”.

Applies statistical mechanics of learning to large-scale recordings over learning with an interesting conceptual framing.

Connects representational similarity, gradient kernels, and universality in a way that inspires new discussion.

Addresses practical issues (e.g., subsampling) that are highly relevant to experimentalists and discusses potential applications (e.g. study of representational drift)

Raises open questions that could guide future theory and experiment.


Weaknesses:

Some technical sections are too detailed for a broad audience and could benefit from more intuitive explanations. The equations lack some definitions, clarity could be improved especially in paragraph line 53-61.

The neuroscience implications remain high-level and would be stronger with concrete experimental examples.

The novelty lies more in synthesis than in new results.

Minor comments: Typos in line 15, line 137

**Score:**

4

**Topic Fit:**

3

---

### Official Review · Reviewer_efAV · 2025-09-15

**Confidence:** 3

**Review:**

**Summary**

The extended abstract provides a cursory overview of summary statistics of learning dynamics, making the case for the utility of summary statistics in terms of compression and predictive capacity. The paper is easy to follow but at time very dense and can profit from being more explicit regarding what type of networks (biological, artificial, spiking) or learning it is arguing for.

**Strengths**
- The paper is generally clearly structured and tackles a relevant problem of both empirical and theoretical dimensions. Obtaining useful permutation-invariant summary statistics of time-varying processes is an important research objective.
- The paper is thematic and provides a checkpoint for a larger perspective / review of summary statistics of learning dynamics.

**Weaknesses**
- The paper makes very smooth conceptual transitions between artificial and biological neural networks, but it remains unclear whether (or is tacitly assumed that) most of the claims apply universally to distributed learning systems in general (maybe the better term than just networks?) or just ANNs. There is also no connection between sections and paragraphs and lots of heavy jargon for a 4-pager.
- The notation is at times a bit hand-waving, terms are not immediately defined after they are introduced, there are no equation numbers making references to terms ever trickier (e.g., the derivative equation / what would be eq. 4 is missing a prime for the second time step $t$)
- The visualizations need to be improved. The labels in Figure 1 are to small and the ones in Figure 1d impossible to read. There is also a disconnect between the caption of Figure 1 and the text. Nowhere does **Section 3** mention dynamic mean field theory, but it seems to be central for understanding Figures 1b and 1c (which are not comprehensible currently). The whole figure should also try to use up maximal horizontal space.

**Score:**

3

**Topic Fit:**

2

---

### Official Review · Reviewer_AZb4 · 2025-09-16
**Unification of learning dynamics in ML with representational similarity in neuroscience.**

**Confidence:** 4

**Review:**

This Perspective outlines how the dynamics of learning in neural networks can be predicted from a small set of so-called “summary statistics of learning” (representational similarity and gradient similarity matrices) using an example of wide/deep neural networks (and shallow networks with high-dimensional data). It then naturally links these statistics to representational similarity matrices in neuroscience and thus links prediction of learning in deep learning with explanations of representation in neuroscience). As I understand it, these results are not new but a synthesis of the literature. In spite of this, the unification of seemingly outwardly distinct concepts from ML and neuro is elegant and a great fit for Unireps.

**Score:**

4

**Topic Fit:**

3